# Facile Fabrication of α-Alumina Hollow Fiber-Supported ZIF-8 Membrane Module and Impurity Effects on Propylene Separation Performance

**DOI:** 10.3390/membranes12101015

**Published:** 2022-10-19

**Authors:** Taewhan Kim, Yeong Jae Kim, Chanjong Yu, Jongbum Kim, Kiwon Eum

**Affiliations:** School of Chemical Engineering, Soongsil University, Seoul 06978, Korea

**Keywords:** membranes, ZIF-8, separation process, hollow fiber

## Abstract

The separation of C_3_ olefin and paraffin, which is essential for the production of propylene, can be facilitated by the ZIF-8 membrane. However, the commercial application of the membrane has not yet been achieved because the fabrication process does not meet industrial regulatory criteria. In this work, we provide a straightforward and cost-effective membrane fabrication technique that permits the rapid synthesis of ZIF-8 hollow fiber membranes. The scalability of the technology was confirmed by the incorporation of three ZIF-8 hollow fiber membranes into a single module using an introduced fiber mounting methodology. The molecular sieving characteristics of the ZIF-8 membrane module on a binary combination of C3 olefin and paraffin (C_3_H_6_/C_3_H_8_ selectivity of 110 and a C_3_H_6_ permeance of 13 GPU) were examined at atmospheric conditions. In addition, the high-pressure performance of these membranes was demonstrated at a 5 bar of equimolar binary feed pressure with a C_3_H_6_/C_3_H_8_ selectivity of 55 and a C_3_H_6_ permeance of 9 GPU due to propylene adsorption site saturation. To further accurately portray the separation performance of the membrane on an actual industrial feed, the effect of impurities (ethylene, ethane, butylene, *i*-butane, and *n*-butane), which can be found in C3 splitters, was investigated and a considerable decrement (~15%) in the propylene permeance upon an interaction with C4 hydrocarbons was confirmed. Finally, the long-term stability of the ZIF-8 membrane was confirmed by continuous operation for almost a month without any loss of its initial performance (C_3_H_6_/C_3_H_8_ separation factor of 110 and a C_3_H_6_ permeance of 13 GPU). From an industrial point of view, this straightforward technique could offer a number of merits such as a short synthesis time, minimal chemical requirements, and excellent reproductivity.

## 1. Introduction

Zeolitic imidazolate frameworks (ZIFs) are a subclass of nanoporous metal organic frameworks (MOFs) composed of organic linkers coordinated to metal ions [1,2]. Their huge pore volume, surface area, and broad range of accessible pore sizes [3] make them very desirable for applications such as adsorption [4,5], catalysis [6,7,8], and membranes [9,10]. In particular, due to its simple synthesis procedure and excellent chemical and thermal stability [11], ZIF-8 is a desirable membrane material [12,13]. In addition, ZIF-8 has shown an excellent pore size range (3–5 Å) [14], desirable for a number of industrially important gas pair separations [13], including C_3_H_6_/C_3_H_8_.

For commercial use, the membrane synthesis approach must fulfill the following key conditions: fewest possible steps in a procedure, high throughput, scalable geometry, competitive fabrication costs, repeatability, and durability [15,16]. Many publications in the literature have aided in the understanding and development of ZIF-8 membranes meeting the criteria mentioned above [17,18,19,20]. For instance, Jeong et al. developed a one-step in situ membrane fabrication process for high-quality ZIF-8 membranes, successfully reducing the procedure step [17]. Briefly, the porous supports were immersed in a metal ion solution and then grown in a ligand solution. When the metal ions and ligand molecules came into contact, the concentration gradients allowed them to migrate in opposite directions (the counter-diffusion concept), forming a ZIF-8 layer at the surface of the porous support. For a high throughput, Tsapatsis et al. adopted the ligand-induced permselectivation technique (an atomic layer deposition of ZnO in a porous support followed by a ligand vapor treatment) to obtain high permeance (1.2 × 10^−7^ mol·m^−2^·s^−1^·Pa^−1^ propylene permeance with 100 propylene/propane selectivity) [18]. Nair et al. developed a polymer hollow fiber-supported ZIF-8 membrane using interfacial microfluidic processing techniques (a two-solvent interfacial approach that could be tuned to achieve positional control over the membrane formation (at the inner and outer surfaces as well as inside the porous hollow fiber), achieving a high volume-to-surface area ratio suitable for a bundle-up) [19]. Mixed matrix membrane (MMM) approaches have also been actively investigated to reduce the cost of the membrane manufacture and improve reproducibility. Recently, Jeong et al. reported on ZIF-8 MMMs using the polymer modification-enabled in situ metal–organic framework (PMMOF) process, successfully suppressing the permeance reduction issue commonly observed in the PMMOF [9]. Finally, a novel approach to improve the mechanical durability of the ZIF-8 membrane was reported via a fast current-driven synthesis (FCDS). In brief, the FCDS induced the ZIF-8 layer to be rapidly developed inside the porous channels of the flexible PP substrates, thereby forming a strong connection between the ZIFs and the support. It was shown that the resulting membrane separation performance remained intact even after severe bending [21].

Despite these excellent recent advances, industrial applications of ZIF-8 membranes have yet to be realized. Rapid thermal deposition (RTD) methods, initially described by Jeong et al. and then improved by Li et al. and Eum et al., have recently shown significant advantages (a simple and rapid synthesis protocol, limited chemical usage, and high reproducibility) [22,23,24]. However, the approach was restricted to a disk-type support. In this study, a facile and scalable ZIF-8 hollow fiber membrane was successfully demonstrated by optimizing the surface characteristics of the alumina fiber and by modifying the RTD procedures. By successfully bundling three ZIF-8 hollow fiber membranes, the scalability of the system was established. The C_3_H_6_/C_3_H_8_ separation performance of the fabricated membrane module was evaluated and its high-pressure applicability was also tested. Furthermore, the influence of impurities on the separation performance was identified for the first time. Finally, we investigated the long-term usage of the membrane.

## 2. Experimental Section

### 2.1. Materials

For the hollow fiber support preparation, α-alumina power was supplied from Baikowski Korea (Seoul, Korea). Polyethersulfone was obtained from Xiamen Keyuan Plastic (Xiamen, China). Polyvinylpyrrolidone (PVP, K30) and N-Methyl-2-pyrrolidone (NMP) were purchased from Daejung Chemicals (Seoul, Korea). For the ZIF-8 membrane fabrication, zinc acetate dihydrate (Zn(OAc)_2_) and 2-methylimidazole (2-mIm) were obtained from Sigma Aldrich (Seoul, Korea). Translucent epoxy (DP-100) was purchased from 3M (Seoul, Korea). Dimethylacetamide (DMAc), methanol (MeOH), and acetic acid (AcOH) were purchased from Daejung Chemicals and the deionized water (DI water) was produced by Direct-Pure Up (Rephile Bioscience, Seoul, Korea).

### 2.2. Methods

#### 2.2.1. α-Alumina Hollow Fiber Preparation 

α-alumina hollow fibers were prepared by dry-jet wet spinning techniques followed by sintering, which can be found elsewhere [25,26]. The detailed spinning conditions are listed in Appendix A. Briefly, a polymer dope solution was made with vigorous stirring for 24 h to ensure a uniform dispersion. The solution was then loaded into a spinning machine and a weak vacuum (10 psia) was maintained overnight to eliminate the majority of the bubbles that had developed within the stirring of the viscous polymer solution. Upon spinning, the raw fiber was solvent-exchanged with methanol followed by hexane, then air dried at 333 K. The fibers were then sintered at 873 K for 3 h, followed by 1723 K (or 1773 K) for 6 h. The temperature ramping rate was maintained at 2 K/min for the entire procedure. The fiber was naturally cooled down.

#### 2.2.2. ZIF-8 α-Alumina Hollow Fiber Membrane Fabrication 

The ZIF-8 membrane fabrication was conducted by preparing two solutions: (1) 1.32 g of Zn(OAc)_2_; and (2) 1 g of 2-mIm dissolved in 15 mL of a DMAc/DI water solution at a ratio of 2:1 (*v*/*v*), respectively. Solution 1 was then added dropwise into Solution 2 and stirred for 15 min. Upon stirring, the solution became murky. The α-alumina fibers were dip-coated into the solution with a linear velocity of 5 mm/min and then immersed for 10 s. A syringe pump was used to improve the reproducibility of the process. The treated α-alumina was placed in an oven preheated to 473 K. The reaction lasted for 15 min before cooling naturally. The ZIF-8 membrane was solvent-exchanged with MeOH for 24 h before drying naturally at an ambient temperature. Before the gas permeation test, the membrane was degassed for 8 h at 393 K.

### 2.3. Characterization 

X-ray diffraction patterns were obtained from a Bruker D2 phaser diffractometer (Billerica, MA, USA) at an ambient temperature using Cu Kα radiation of λ = 0.154 nm and a scanning range of 5–40° 2θ. Surface and cross-section SEM images were collected with a ZEISS GeminiSEM 300 (Oberkochen, Germany). All membrane samples were pretreated with sputter-coated Pt (Q150R Plus-Rotary Pumped Coater, East Sussex, UK). The EDX elemental analysis was performed with a SEM-attached Bruker XFlash 6I30 (Billerica, MA, USA). The pore size of the α-alumina was obtained from a Micrometrics AutoPore IV 9500 (Norcross, GA, USA).

### 2.4. Gas Permeation Test 

The Wicke–Kallenbach (W–K) approach was used to test the C_3_H_6_/C_3_H_8_ binary gas mixture (Figure 1). As the feed side, an equimolar C_3_H_6_/C_3_H_8_ binary mixture was utilized. An additional line was attached to the feed side for the tertiary mixture tests (C_3_H_6_/C_3_H_8_ and impurities). Argon was employed as a permeate stream sweep gas. The flow rates of the C_3_H_6_/C_3_H_8_ mixture and the argon gas were all set to 20 SCCM, respectively. In the case of the tertiary mixture test, the flow rate of the impurities was set to 0.3 SCCM so that the molar ratio of the C_3_H_6_/C_3_H_8_/impurity was 49.3:49.3:1.4. A back-pressure regulator was used to control the transmembrane pressure. A forced convection oven regulated the temperature of the membrane module (Yamato DKN312C, Tokyo, Japan). Online gas chromatography was used to examine the composition of the permeate side stream (Agilent GC 6890N, Santa Clara, CA, USA). The system was stabilized for at least 4 h before each measurement and the measurement was repeated numerous times until it achieved a steady state value. The binary or tertiary gas permeation calculation methodology is provided in the Appendix A.

## 3. Results and Discussion

Our simple and facile strategy for the formation of scalable ZIF-8 hollow fiber membranes is illustrated in Figure 2. Briefly, a 5 cm long bare α-alumina hollow fiber was positioned vertically inside the vial that was attached to the syringe pump. The RTD solution (Zn^2+^ + 2-mIm in 2:1 (*v*/*v*) DMAc/H_2_O solution) was then injected with a syringe pump at a linear velocity of 5 mm/min into the bottom of the vial. After the solution had dwelled at the top for 10 s, it was withdrawn at the same rate of 5 mm/min. During this period, a portion of the precursor-containing solution was absorbed through the shell side of the hollow fiber by osmotic pressure. After that, the support was heated to a high temperature (200 °C) so that the solvent could evaporate. This caused the system to become oversaturated, which was when ZIF-8 started to form and quickly crystallize. 

As the RTD techniques relies on the osmosis of a precursor-containing solution via microscopic pores in the support, the surface characteristics (e.g., the morphology, pore size, and roughness) of the support are critical. If the pore size is too small, a sufficient amount of the solvent may not be absorbed into the support, resulting in membrane defects. In the event of a support with excessively huge pores, the thickness of the membrane may be unnecessarily increased, owing to deep solvent penetration, resulting in a loss of the membrane performance. In this work, a dry-jet wet fiber spinning technique was used for the α-alumina hollow fiber support preparation and the pore size of the support could be adjusted by altering the calcination temperature. Top-view SEM images of the homemade α-alumina hollow fibers calcined at 1773 K and 1723 K are shown in Figure 3a,b. For substrates that were calcined at temperatures of 1773 K, the alumina particles were overly sintered to the point where the pores were almost impossible to find. A low N_2_ permeance of 250 GPU also supported the presence of small pores. For the α-alumina substrates calcined at 1723 K, visible pores with a N_2_ permeance of 80,000 GPU were observed. The mercury porosimeter result also confirmed the presence of pores between 200 and 900 nm in diameter (Appendix A). Additionally, when the hollow fiber was calcined below 1700 K, its mechanical strength was significantly reduced, making it unsuitable as a membrane support. 

The RTD membrane growth technique was used for both the 1773 K- and 1723 K-calcined alumina supports; the top-view SEM images of both membranes are shown in Figure 4a,b. The limited penetration of the RTD precursor solution into the 1773 K-calcined α-alumina support (Figure 4a) resulted in no apparent surface morphology changes and the formation of severely defective layers (C_3_H_6_/C_3_H_8_ selectivity = 1, equimolar binary C_3_H_6_/C_3_H_8_ gas mixture at 1 bar and RT) was confirmed; however, the surface of the 1723 K-calcined support (Figure 4b) was consistently covered with a ZIF-8 layer and no obvious defects such as pinholes or cracks were detected. The XRD pattern (Figure 4c) of the membrane (1773 K-calcined α-alumina support) also confirmed that a ZIF-8 layer was formed. Based on the cross-sectional SEM image of the ZIF-8 membrane (1773 K-calcined α-alumina support), the thickness of the ZIF-8 layer was measured at 2.8 μm ± 0.4 μm (averaged from 10 different location measurements) and a huge amount of the ZIF-8 layer was infiltrated inside the porous support (Figure 4d). The EDX mapping of the zinc along the radial direction of the hollow fiber verified the localization of the ZIF-8 layer on the outer surface of the fiber as well as the penetration of the ZIF-8 layer into the support. This penetrated ZIF-8 layer had certain downsides in terms of permeance, but it may have been advantageous in terms of mechanical stability. The ZIF-8 layer was still intact after 5 min of sonication.

The three as-made ZIF-8 hollow fiber membranes were then combined into a single module to demonstrate the scalability of the technology. A well-defined and reliable fiber mounting methodology should be established in order to prevent the leakage of the feed gas at the connection between the fiber and the module. In particular, when multiple fibers are bundled, sealing between the fibers on the end side is crucial. Our membrane module preparation protocol is illustrated in Figure 5. Briefly, the membrane module was first prepared by inserting three fibers into the permeation testing module and sealing the ends with epoxy. After the epoxy had cured for 24 h, a plastic tube was inserted through the outside wall of the module. The module was then positioned vertically and additional epoxy was supplied via the tube, allowing gravity to complete the filling of the plastic tube. The procedure was subsequently repeated on the opposite side of the module. The epoxy was then removed by gently twisting the plastic tubes, which revealed the fiber cross-sections on the outside of the module, confirming that they were made with no defects, perfectly connected to the epoxy, and not crumpled or distorted in any way. 

The membrane modules were subjected to equimolar binary C_3_H_6_/C_3_H_8_ feed pressures ranging from 1 to 5 bars at 273 K and the Wicke–Kallenbach technique was used to evaluate the performance (Figure 6). Three independently prepared membrane modules were tested and averaged. A C_3_H_6_/C_3_H_8_ selectivity of 110 and a C_3_H_6_ permeance of 13 GPU were obtained at 1 bar of feed pressure, indicating solid evidence for molecular sieving behavior. The transmembrane pressure was then raised by increasing the feed pressure whilst the permeate pressure remained constant at 1 bar. The separation factor dropped as the transmembrane pressure increased (owing to a decrease in the C_3_H_6_ permeance), yet a high selectivity of 55 was still attained at a feed pressure of 5 bar. Pressure-induced decreases in the C_3_H_6_ permeance and a persistently low value for the C_3_H_8_ permeance (about 0.1 GPU) defied an explanation by the existence of high-pressure defects, which would have otherwise led to substantial increases in the permeance of both components of the mixture. On the basis of the basic properties of transport in nanoporous materials, an additional explanation was proposed. At a high pressure, the adsorption site saturation may have resulted in a decrease in the propylene permeance (permeability coefficient = solubility coefficient x diffusion coefficient). It is worth noting that, despite the reduction in the C_3_H_6_ permeance, increasing the feed pressure driving force resulted in a significant increase in the C_3_H_6_ flux (by around a factor of 3 at ΔP = 4 bar) in comparison with the ΔP = 0 case.

Other than the C_3_H_6_/C_3_H_8_ mixture, the industrial input stream comprised a substantial number of impurities. Table 1 depicts the multicomponent mixture departing the fluid catalytic cracking (FCC) unit and entering the C3 splitter. We examined how the impurities impacted the C_3_H_6_/C_3_H_8_ separation performance on the ZIF-8 membrane. A tertiary gas mixture was prepared by mixing an equimolar C_3_H_6_/C_3_H_8_ mixture stream with an impurities stream. The composition of impurities was set to 1.4% in all cases (the remaining 98.6% being a C_3_H_6_/C_3_H_8_ mixture). The percentage of the propylene permeance decrease relative to the equimolar binary C_3_H_6_/C_3_H_8_ mixture result was calculated, as shown in Figure 7. The impurity effects could be divided into two categories: impacts on (1) C2; and (2) C4 hydrocarbons. The C2 hydrocarbons had minor effects on the C_3_H_6_ permeance in general. With the C4 hydrocarbons, however, there was a ~15% drop in the C_3_H_6_ permeance. This trend of the C_3_H_6_ permeance reduction was found to correlate with the adsorption affinity of the impurities for ZIF-8. The Henry constant of ethylene, ethane, propane, propylene, butylene, *i*-butane, and *n*-butane in ZIF-8 can be found in Table 1. It was noted that the Henry constant of propylene for ZIF-8 was 10.9 mmol·g^−1^·bar^−1^. In cases where the Henry constant of impurities was much greater than that of propylene (e.g., *n*-butane, butylene, and *i*-butane), propylene competed with the impurities (albeit in low concentrations) to be adsorbed on the ZIF-8 surface. A decrease in the propylene adsorption concentration on the surface of the membrane decreased the permeance of the membrane. In conclusion, in the event that an impurity exhibited substantial adsorption with ZIF-8, it was crucial to eliminate it in advance or maintain its concentration at a very low level.

The solvent reusability was demonstrated by repeatedly fabricating the ZIF-8 hollow fiber membranes using the same mother RTD solution. As shown in Table 2, the mother RTD solution could be reused at least 5 times (1st–5th fabrication) without compromising the membrane performance. For the 6th fabrication, the solvent was maintained on the shelves for 1 month and the solution could still be used for excellent quality membrane fabrication. Finally, Figure 8 shows the results of a ZIF-8 hollow fiber membrane module that was continuously operated for 30 days at 25 °C and 1 bar feed pressure with an equimolar binary C_3_H_6_/C_3_H_8_ feed. Initially, the membrane module showed a C_3_H_6_/C_3_H_8_ selectivity of 110 and a C_3_H_6_ permeance of 13 GPU; both of these values remained constant after 30 days of operation.

## 4. Conclusions

In conclusion, a simple and facile strategy for the formation of scalable ZIF-8 hollow fiber membranes was illustrated upon a modification of the rapid thermal deposition (RTD) technique. The surface characteristics of the fiber support were carefully adjusted to provide a more open and uniform surface pore structure that enabled uniform RTD solvent penetration. In addition, it was demonstrated that the same mother RTD solution could be utilized several times (at least five times) within a month to produce good-quality ZIF-8 membranes. The membrane thickness was estimated to be 2.8 μm and the polycrystalline layer remained intact after 5 min of sonication, proving a good mechanical stability. Three as-made ZIF-8 fibers were bundled into single modules and the molecular sieving properties were investigated with C_3_H_6_ and C_3_H_8_ binary mixed gas pair (C_3_H_6_ permeance and separation factor of 13 GPU and 110, respectively). At high pressures of up to 4 bars, the separation factor was maintained over 50, achieving 3 times the flux of atmospheric conditions. The performance of the membrane was further examined in the presence of hydrocarbon impurities and it was shown that C4 hydrocarbons significantly affected the propylene permeance (a decrease of ~15%) by competitive adsorption. Finally, even after more than a month of continuous operation, the membrane exhibited no performance decline. 

## Figures and Tables

**Figure 1 membranes-12-01015-f001:**
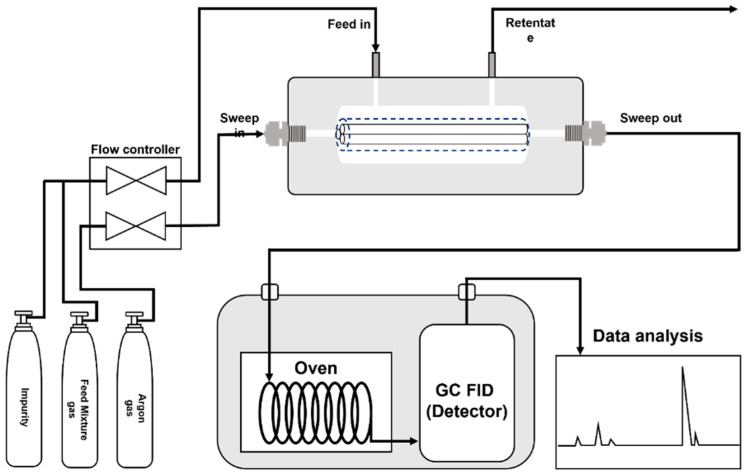
Schematic illustration of the experimental setup for the binary or tertiary gas mixture permeation setup.

**Figure 2 membranes-12-01015-f002:**
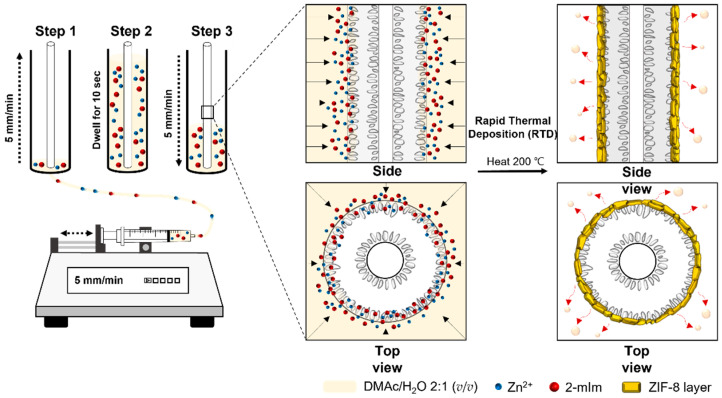
Schematic illustration of a modified rapid thermal deposition approach for the fabrication of α-alumina hollow fiber-supported ZIF-8 membranes.

**Figure 3 membranes-12-01015-f003:**
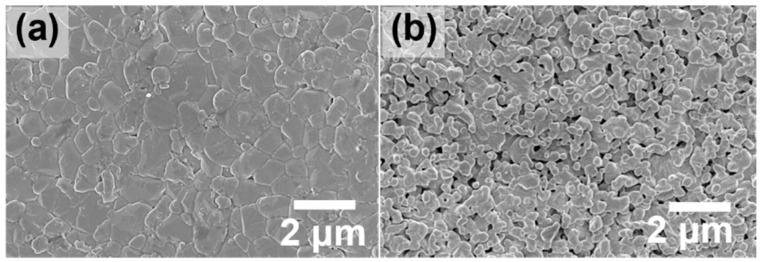
Top-view SEM images of α-alumina support calcined at (**a**) 1773 K and (**b**) 1723 K.

**Figure 4 membranes-12-01015-f004:**
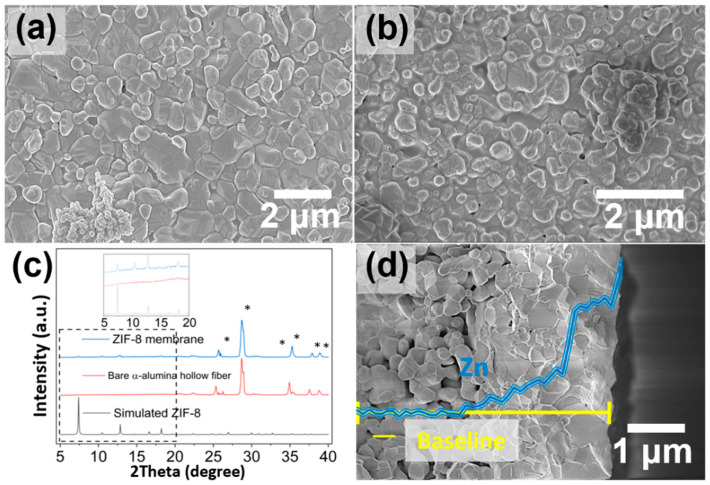
Top-view SEM images of ZIF-8 membrane grown on (**a**) 1773 K- and (**b**) 1723 K-calcined α-alumina hollow fiber supports, (**c**) XRD patterns (* represents XRD diffraction peaks of α-alumina support and clay from XRD sample preparation), and (**d**) cross-sectional SEM images of ZIF-8 membrane grown on 1773K-calcined α-alumina support (EDX elemental line maps of Zn^2+^ (blue) across the radial direction of the fiber support showing the localization of the ZIF-8 membrane to the outer side of the fiber).

**Figure 5 membranes-12-01015-f005:**
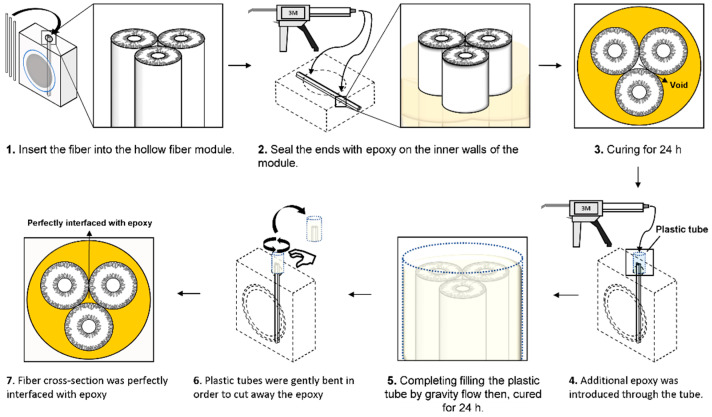
Schematic illustration depicting the installation of three ZIF-8 hollow fiber modules.

**Figure 6 membranes-12-01015-f006:**
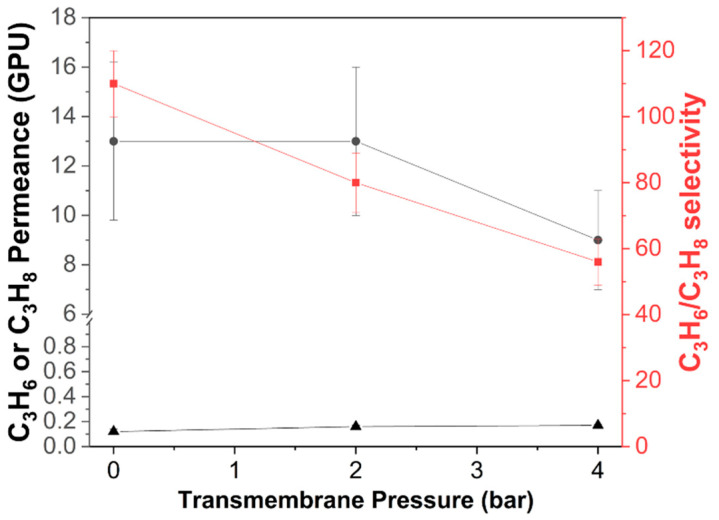
Equimolar binary C_3_H_6_/C_3_H_8_ separation properties of the ZIF-8 hollow fiber membrane module (three ZIF-8 hollow fibers were bundled) as a function of transmembrane pressure (0–4 bar) at 298 K. The error bar was obtained with three independently prepared membrane modules.

**Figure 7 membranes-12-01015-f007:**
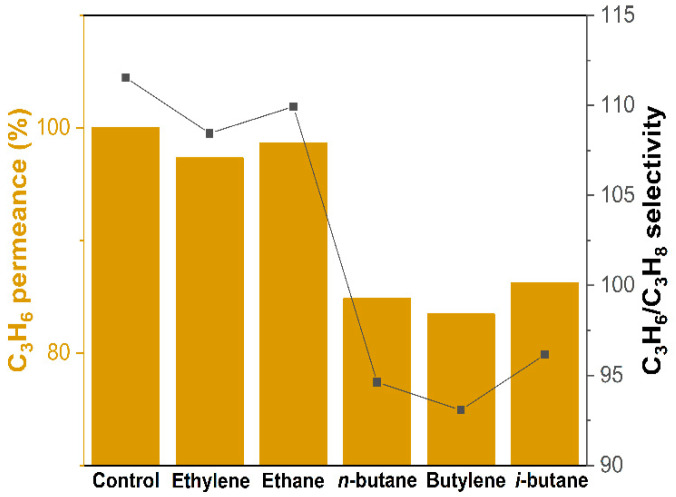
Tertiary gas mixture C_3_H_6_/C_3_H_8_ impurity (49.3:49.3:1.4) separation properties of the ZIF-8 hollow fiber membrane. The impurities (ethylene, ethane, *n*-butane, butylene, and isobutane) that can be found in an industrial C3 splitter feed stream were tested.

**Figure 8 membranes-12-01015-f008:**
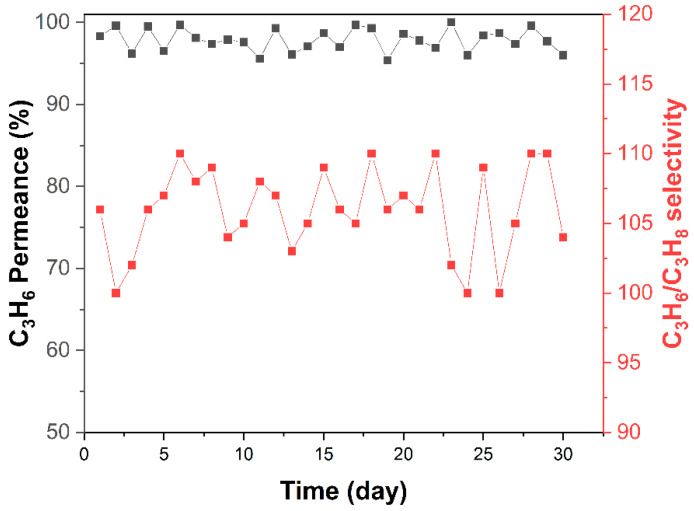
Performance of a ZIF-8 hollow fiber membrane operated continuously for a month under an equimolar C_3_H_6_/C_3_H_8_ mixture feed at 25 °C and 1 bar feed pressure. The percentage propylene permeance change relative to the day 0 value was calculated and demonstrated.

**Table 1 membranes-12-01015-t001:** Component and percent of mixture exiting the FCC unit and entering the C3 splitter. The Henry constant of each component at 35 °C was obtained from [27].

Component	Ethylene	Ethane	Propylene	Propane	Butylene	*i*-Butane	*n*-Butane
Percent (%)	0.1	1.1	68.7	27.8	1.1	0.8	0.5
Henry constant (mmol·g^−1^·bar^−1^)	1.18	2.15	10.9	14.6	95.9	97.4	66.1

**Table 2 membranes-12-01015-t002:** RTD precursor solution reusability test.

Number of Repeat	C_3_H_6_ Permeance (GPU)	C_3_H_8_ Permeance (GPU)	C_3_H_6_/C_3_H_8_ Selectivity
1st fabrication	13.2	0.12	110
2nd fabrication	12.6	0.13	97
3rd fabrication	14.6	0.11	133
4th fabrication	12.1	0.12	101
5th fabrication	13.6	0.13	105
6th fabrication	13.8	0.12	115

## Data Availability

The data used to support the findings of this study are available from the corresponding author upon request.

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
