# Peer review of "Facile Fabrication of α-Alumina Hollow Fiber-Supported ZIF-8 Membrane Module and Impurity Effects on Propylene Separation Performance"

_membranes, 2022, doi:10.3390/membranes12101015_

Round 1

Reviewer 1 Report

In this manuscript, the rapid thermal deposition (RTD) technique was modified for the facile and scalable preparation of hollow fiber supported ZIF-8 membrane. The reproducibility of the modification strategy as well as the pressure resistance, long-term stability and impurity resistance of the prepared membrane were evaluated and discussed. Although the C3H6/C3H8 separation performance and pressure resistance of the membrane do not match the state of the art, this straightforward synthesis strategy still holds promise considering its simplicity, efficiency, and good reproducibility. Therefore, I suggest that it can be considered for acceptance after major revision.

1. Authors should double-check the content of the manuscript to avoid superficial problems like repetition of words (e.g., 2-mIm dissolved in dissolved in 15 ml of…) and format of corner markers (e.g., 1.2×10-7 mol·m-2·s-1·Pa-1 and DMAc/H2O solution).

2. The specific values of the feed gas flow rate ( including C3H6, C3H8 and impurity gases ) should be given in the gas permeation test section.

3. In addition to excessive membrane thickness (~2.8 μm) and embedded DMAc in the cavity of ZIF-8, over-sintered α-Al2O3 hollow fiber at 1773 K with barely observable pores are mainly responsible for the relatively low C3H6 permeance (~13 GPU at RT and 1 bar) of prepared ZIF-8 membrane. Therefore, necessary explanations and data (e.g., pore size, C3H6 permeance and cross-sectional SEM image of bare substrate) are indispensable to elaborate the above issue clearly.

4. For RTD process, trapped precursor solution will crystallize from inner side to outer side of the substrate. It is suggested that the authors should give cross-sectional EDXS mapping to reveal the infiltration depth of the ZIF-8 membrane.

5. The C3H6 permeance decreased with increasing feed pressure due to the saturation of the adsorption sites at high pressure. If so, what is the reason for the almost constant C3H8 permeance? And what causes the decline of C3H6/C3H8 selectivity at elevated pressure

6. As far as I know, some authors have reported the C3H6/C3H8 separation performance of ZIF-8 membrane in the presence of impurities. It is not objective for the author to consider himself the first to engage in such research.

Author Response

We have provide the point-by-point response to the reviewer`s comments below. Thanks for all the good comments and the suggetion. 

Reviewer

Comments

Responses

Reviewer #1

Authors should double-check the content of the manuscript to avoid superficial problems like repetition of words (e.g., 2-mIm dissolved in dissolved in 15 ml of…) and format of corner markers (e.g., 1.2×10-7 mol·m-2·s-1·Pa-1 and DMAc/H2O solution).

Thanks for your corrections of typos and errors. Now the manuscript has been updated accordingly.

The specific values of the feed gas flow rate (including C3H6, C3H8 and impurity gases) should be given in the gas permeation test section.

The flow rate of the C3H6, C3H8, and impurities are set to 10 SCCM, respectively. The manuscript has been updated accordingly.

In addition to excessive membrane thickness (~2.8 μm) and embedded DMAc in the cavity of ZIF-8, over-sintered α-Al2O3 hollow fiber at 1773 K with barely observable pores are mainly responsible for the relatively low C3H6 permeance (~13 GPU at RT and 1 bar) of prepared ZIF-8 membrane. Therefore, necessary explanations and data (e.g., pore size, C3H6 permeance and cross-sectional SEM image of bare substrate) are indispensable to elaborate the above issue clearly.

For better clarity, We have rewritten the Figure 4 caption and the corresponding section of the manuscript. All the changes were highlighted in the revised manuscript.

Briefly, the ZIF-8 layer grown on the surface of the 1773 K calcined alumina support turned out to be defective one due to insufficient infiltration of the precursor-containing solution. No further investigation was performed with the membrane since it shows no molecular sieving properties upon C3 olefin/paraffin mixture. (C3H6/C3H8 selectivity = 1 and propylene permeance is ~200 GPU)

The 1723K calcined hollow fiber supported ZIf-8 membrane shows good C3 olefin/paraffin separation performance and the corresponding suggested characterization were performed.

For RTD process, trapped precursor solution will crystallize from inner side to outer side of the substrate. It is suggested that the authors should give cross-sectional EDXS mapping to reveal the infiltration depth of the ZIF-8 membrane.

Cross-sectional EDX line mapping result of ZIF-8 hollow fiber membrane can be found in Figure 3 (d).

The C3H6 permeance decreased with increasing feed pressure due to the saturation of the adsorption sites at high pressure. If so, what is the reason for the almost constant C3H8 permeance? And what causes the decline of C3H6/C3H8 selectivity at elevated pressure

For propane, the activation energy of diffusion is greater than the heat of adsorption; thus, it is a diffusion-dominated procedure with minimal adsorption contribution. Please see below for your reference. (10.1016/j.memsci.2011.11.024)

At high pressure, the decrease in propylene permeance results in a decrease in selectivity. (Please note the propane permeance does not changed)

Reviewer 2 Report

The paper is very interesting however it requires major revision before being accepted.

particularly

1. the introduction should be improved, review important papers and make clearer what the novelty of the paper is compared to the knowledge available in literature

2. better description of the techniques used and results

3. most of the results should also show the error bars and the errors in the measurements need to be better discussed

4. the English should be checked for typos.

Author Response

We have provide the point-by-point response to the reviewer`s comments below. Thanks for all the good comments and the suggetion.

Reviewer

Comments

Responses

Reviewer 2

The introduction should be improved, review important papers and make clearer what the novelty of the paper is compared to the knowledge available in literature

The introduction part was partially rewritten to introduce the important recent related works and highlight our findings.

better description of the techniques used and results.

Detailed descriptions and techniques were included in the revised manuscript. All the changes were highlighted in the revised manuscript.

most of the results should also show the error bars and the errors in the measurements need to be better discussed

The method for collecting the error Bar was described in detail in the revised manuscript.

the English should be checked for typos.

Typos and errors were corrected in the revised manuscript.

Reviewer 3 Report

1.     The authors need to rewrite the abstract quantitatively; they should mention salient quantitative findings in the abstract.

2.     The author should extract the information from each reference. They need to avoid lumped references where it is applicable. Such examples are [1-4], [7-9], and [11-14], etc.

3.     In Table 1, in the supplementary information, the authors did not mention the “take-up rate.”

4.     The authors should move Figure S1 (in section 2.4) and Figure S2 (in the appropriate section) to the main article for better understanding.

5.     Why authors did not perform the thermal analysis (TGA or DSC) for the developed membranes.

6.     At what pressure authors performed the tertiary gas mixture analysis?

7.     The authors explained the figures well but need to support all the findings with well-cited recent literature.

Author Response

We have provide the point-by-point response to the reviewer`s comments below. Thanks for all the good comments and the suggetion.

Reviewer

Comments

Responses

Reviewer #2

The authors need to rewrite the abstract quantitatively; they should mention salient quantitative findings in the abstract.

The abstract has been updated to include the quantitative values. All the changes were highlighted in the revised manuscript.

The author should extract the information from each reference. They need to avoid lumped references where it is applicable. Such examples are [1-4], [7-9], and [11-14], etc.

The introduction part was partially rewritten, and the reference was revised to indicate the specific information, avoiding lumping.

In Table 1, in the supplementary information, the authors did not mention the “take-up rate.”

It is intentionally /* foot noted and commented on the caption.

For α-alumina hollow fiber fabrication, the raw fiber was collected from the bottom of the quenching bath instead of using the take-up drum.

The authors should move Figure S1 (in section 2.4) and Figure S2 (in the appropriate section) to the main article for better understanding.

Thanks for the suggestion. The Figure S1 was relocated to the main manuscript.

Why authors did not perform the thermal analysis (TGA or DSC) for the developed membranes.

Typically, TGA can be used to characterize the loading (wt%) of ZIF-8 particle in mixed-matrix membrane. Thickness of the polycrystalline membrane can rather be characterized with SEM, TEM or EDX.

At what pressure authors performed the tertiary gas mixture analysis?

Thank you for informing us of omitted information. Here we used 1 bar for tertiary feed gas mixture.

The authors explained the figures well but need to support all the findings with well-cited recent literature.

Well cited recent articles are included in revised manuscript.

Round 2

Reviewer 1 Report

The authors have fully addressed the concerns raised. It is therefore commended for acceptance in the current form.

Reviewer 2 Report

the authors made the changes required

Reviewer 3 Report

The authors addressed all the comments satisfactorily. The paper can be accepted for publication after a minor language check.